# Relationship between Inflammation and Vasospastic Angina

**DOI:** 10.3390/medicina59020318

**Published:** 2023-02-09

**Authors:** Ming-Yow Hung, Ming-Jui Hung

**Affiliations:** 1Division of Cardiology, Department of Internal Medicine, School of Medicine, College of Medicine, Taipei Medical University, Taipei 110, Taiwan; 2Taipei Heart Institute, Taipei Medical University, Taipei 110, Taiwan; 3Division of Cardiology, Department of Internal Medicine, Shuang Ho Hospital, Taipei Medical University, New Taipei City 23561, Taiwan; 4Section of Cardiovascular Imaging, Division of Cardiology, Department of Internal Medicine, Chang Gung Memorial Hospital at Keelung, Chang Gung University College of Medicine, Taoyuan 333, Taiwan

**Keywords:** angina, coronary artery disease, inflammation, vasospasm

## Abstract

Coronary artery spasm (CAS) is a dynamic coronary stenosis causing vasospastic angina (VSA). However, VSA is a potentially lethal medical condition with multiple presentations, including sudden cardiac death. Despite investigations to explore its pathogenesis, no single mechanism has been found to explain the entire process of VSA occurrence. The roles of elevated local and systemic inflammation have been increasingly recognized in VSA. Treatment strategies to decrease local and systemic inflammation deserve further investigation.

## 1. History of Coronary Artery Spasm (CAS)

Prinzmetal and colleagues observed an atypical angina occurring at rest associated with an elevated ST segment on electrocardiograms transiently in patients with atherosclerotic coronary artery disease (CAD) [1]. The angina would have been due to a transient decrease in coronary blood flow, since, at rest, cardiac work is not increased. Subsequently, the term “variant angina” was suggested by Prinzmetal et al. in 1959, and they suggested that CAS was the cause because it was relieved immediately after administrating nitroglycerin. In the 1970s, variant angina was found to be caused by CAS, which was confirmed by coronary angiography. CAS can potentially occur at the site of atherosclerotic CAD [2] or diffuse spastic changes in angiographically normal coronary arteries. As a result, the investigators termed it a “variant of the variant” [3] or “vasospastic angina (VSA)” [4]. The majority of CAS cases are accompanied by ST-segment depression or T-wave changes instead of ST-segment elevation [5,6,7]. Therefore, the term “VSA” is a broader term to represent CAS-induced angina, irrespective of electrocardiographic manifestations. The term “variant angina” is usually expressed as CAS-induced angina associated with concurrent ST-segment elevation transiently on electrocardiogram. Recently, a Japanese guideline development by the Japanese Circulation Society has suggested that variant angina is a type of VSA [4]. The CAS experts in the Joint Working Groups in Japan [4] proposed using the term “VSA” to represent coronary vasomotor-disorder-related angina and this concept has been widely accepted [8].

## 2. How to Diagnose and Treat VSA

VSA differs from typical angina in its pathogenesis, although the exact pathophysiology of VSA is not clear at present. VSA usually occurs during resting status, especially in the night and early morning, but we found that some patients may have angina with ST-segment deviations during exercise [9]. We suggested that the spastic coronary arteries are abnormal, as the dilator response to exercise is not adequate as it would be in normal coronary arteries. There are variations in the occurrence of VSA, i.e., daily, weekly, monthly, and circadian [10]. Many cardiologists have found that CAS can cause stable angina, acute coronary syndrome, syncope, heart failure, cardiac arrhythmias, and even sudden cardiac death [4,11]. Therefore, it is crucial to identify CAS as the underlying cause of a cardiovascular event because the treatment options will be different according to the diagnosis, i.e., pharmacological treatment first for CAS-induced and pharmacological treatment plus coronary intervention for atherosclerotic coronary artery stenoses. As a matter of fact, a correct diagnosis leads to correct treatments, and this logic of causality is the core value of clinical medicine. Recently, guidelines developed by the European Society of Cardiology for the management of survivors of sudden cardiac death have suggested that the diagnosis of CAS-induced sudden cardiac death may be considered [12]. This shows that cardiovascular events caused by CAS have been paid more and more attention.

Yasue et al. [10] found that the culprit coronary artery was patent in 17.9% of patients with acute myocardial infarction, which is similar to our report of 12% [13]. In our report, infarct-related CAS could be provoked in 95% of acute myocardial infarction patients, which suggests that a transient process of spasm and/or thrombus resolution occurs in these patients. CAS causes the formation of intracoronary thrombus [14], suggesting that CAS is a cause of acute myocardial infarction.

In the 1980s–2000s, VSA was reported to have a higher prevalence in the Japanese population compared with the western population [15,16]. Subsequently, the prevalence rates of VSA in Taiwan [17] and Korea [18] were reported to be similar to Japan [19]. The incidence of CAS provocation in our study (54%) [17] and that of Kim et al. (48%) [18] was higher than that of Bertrand et al. (12.3%) [15]. Furthermore, the rate of inducible multi-vessel spasms (2- and 3-vessel spasms) in our study (19%) [17] was higher than in Bertrand et al. (7.5%) [15]. However, recent European studies found that VSA is not as uncommon as previously thought in white patients with angina pectoris and myocardial ischemia with unobstructive coronary arteries [20,21]. There were no differences in patterns (more diffuse spasms and similar 2- or 3- vessel spasms) of CAS in these studies; however, higher proportions of males and smoking history were noted in the Japanese population [18]. Recent studies evaluated CAS systematically in patients who had angina or myocardial ischemia without obstructive coronary arteries. The authors found that epicardial CAS and microvascular spasm are important causes in these patients, indicating that coronary vasomotor testing should be undergone for patients with angina and no obstructive coronary arteries. The awareness of assessment for coronary vasomotor disorder has formed a consensus in the communities of cardiology and cardiovascular intervention. The probable underdiagnosis of VSA in the world, especially in the western population, should not be overlooked. In fact, the actual frequency of VSA occurrences is not easy to define because occurrences of VSA tend to fluctuate and are not necessarily symptomatic. In other words, silent ischemic CAS is a possible clinical entity. Therefore, ethnic heterogeneities in VSA require further research. No angiographic CAD can be found in one-fourth of patients with acute coronary syndrome [22,23]. CAS can be provoked in around 50% of these patients. If the electrocardiographic ST-segment changes are normalized after initial management (i.e., oxygen, aspirin, and nitroglycerin), CAS is a major factor contributing to the acute coronary syndrome. Emergency cardiac catheterization and coronary intervention is not strictly necessary under this circumstance. However, frequent attacks of VSA are a strong indication for emergency cardiac catheterization to evaluate the underlying coronary artery pathology and to perform coronary intervention if necessary. Therefore, follow-up electrocardiograms are crucial in diagnosing CAS-related acute coronary syndrome.

It is necessary to evaluate coronary function and to clarify the role of CAS in angina pectoris. Using intracoronary provocative testing to perform coronary function testing is needed and relatively safe [24], especially in patients with ischemia and nonobstructive coronary arteries [25]. Ergonovine maleate, methylergonovine maleate, and acetylcholine have been used effectively to induce CAS. The intracoronary route of ergonovine administration, with a step-wise dosing of 1, 5, 10, and 30 μg and a 3-min interval between doses, has high sensitivity and specificity in inducing CAS [26,27]. Usually, the right coronary artery is evaluated first, then the left coronary artery. CAS is defined as a >70% reduction in the coronary arterial luminal diameter associated with chest pain and/or electrocardiographic ST-segment and T-wave changes during provocation testing [13,28]. A positive provocative test as a decrease of >90% in the coronary arterial diameter associated with chest pain and/or electrocardiographic ST-segment deviations during the provocation testing has been suggested by a Japanese guideline [4]. However, Yasue et al. [29] suggested that myocardial ischemia could be caused by reduced coronary blood flow for long enough. Therefore, a definition of coronary artery lumen reduction seems to be of no absolute necessity. The core of a positive CAS provocation test result is concurrent angina and/or ischemic electrocardiographic changes during testing. Therefore, simultaneous patient symptom inquiry and electrocardiographic monitoring are absolutely necessary. Pre-testing, liquid nitroglycerin must be well prepared, and 50–600 μg of intracoronary nitroglycerin is administered once a CAS has been diagnosed. Certainly, intracoronary ergonovine administration must be stopped before intracoronary nitroglycerin administration. Only methylergonovine maleate is available in Taiwan; therefore, it was used in our prior studies with the intracoronary dose protocol the same as for ergonovine maleate. This procedure was safe with low complication rates [24,30]. Reported complications of intracoronary provocative testing for CAS include angina, atrial or ventricular arrhythmias, hypotension, nausea, vomiting, and flushing. There have been no reports regarding procedure-related mortality or myocardial infarction. It is recommended that CAS provocative testing should be undergone in a cardiac catheterization because of possible fatal or non-fatal arrhythmias occurring during testing. Therefore, it is not advisable not to undergo intracoronary provocative testing for fear of complications, as with all cardiac interventions. A complete and correct diagnosis should be made for the patient who has angina and no obstructive coronary arteries as long as there is detailed and complete preparation before testing. Theoretically, the diagnosis of VSA should be made according to the intracoronary provocation testing result; however, it is not practical to undertake intracoronary provocation testing immediately after an attack of VSA in every patient. However, some clues are more likely to reflect VSA: (1) chest pain occurs in resting status, especially at night and in the early morning; (2) chest pain is associated with concurrent electrocardiographic ST-segment and T-wave changes; (3) chest pain is quickly relieved by nitroglycerin in any form. Even so, it is still advisable to undertake intracoronary provocative testing if there is no contraindication. 

In addition to intracoronary stress testing, other non-invasive stress modalities have been used to diagnose VSA, such as hyperventilation [31] and stress echocardiography using either cold-pressor testing or intravenous ergonovine testing [32,33]. In 1999, Nakao et al. [31] studied 206 angiographically confirmed CAS patients (spasm group) and 183 non-angina and non-angiographically-inducible CAS patients (non-spasm group) using vigorous hyperventilation for 6 min in the early morning. Of these 206 patients, 127 had positive electrocardiographic responses to the test; however, all negative responses were noted in the non-spasm group. As a result, the sensitivity and specificity of hyperventilation testing for diagnosing CAS were 62% and 100%, respectively. The postulated mechanism is that respiratory alkalosis induced by hyperventilation enhances Na-H exchange followed by Na-Ca exchange, subsequently causing increased intracellular calcium concentration. In 2001, Hirano et al. [32] reported 2-dimensional echocardiographic stress testing to evaluate CAS. The stress testing includes hyperventilation for 6 min, followed by cold water pressor stress for 2 min. The whole process was closely monitored by continuous electrocardiograms and echocardiograms. The sensitivity, specificity, and diagnostic accuracy of this stress testing protocol for detecting CAS were 48%, 100%, and 60%, respectively. These results mean that these tests are specific for CAS. In other words, CAS truly exists when angina occurs after hyperventilation. In 2005, Song et al. [33] reported the role of intravenous ergonovine stress echocardiography in the diagnosis of CAS. The positive rate was 8.6% for detecting CAS, and no procedure-related mortality or myocardial infarction was noted. Based on the above studies, it is suggested that these non-invasive modalities are alternative methods in diagnosing VSA if there are contraindications to undergoing invasive coronary angiography, and these are suggested to be performed by experienced physicians. Although diagnosis of VSA can be made invasively and non-invasively, contraindications to performing these tests still exist and need attention. Absolute contraindications to undergoing CAS provocation testing include severe left ventricular dysfunction, moderate to severe aortic stenosis, high-grade left main coronary artery stenosis, severe hypertension (systolic blood pressure > 180 mmHg), and pregnancy [34]. Relative contraindications include significant coronary artery disease, recent myocardial infarction, uncontrolled or unstable angina, and uncontrolled ventricular arrhythmia. Based on the above literature reviews, taking a thorough medical history and a follow-up series of electrocardiographic ST-segment and T-wave changes are the bases for diagnosing VSA.

Calcium antagonists are the first-line therapy in the treatment of VSA [4,10,35]. Calcium antagonists are suggested to be given before bedtime at night because VSA frequently occurs between midnight and early morning. Furthermore, the doses of calcium antagonists for VSA are not the same as those for treating hypertension; a larger dose of calcium antagonist is usually needed, e.g., diltiazem 240–360 mg/day. Controlling VSA may occasionally require two distinct chemical classes of calcium antagonists, i.e., dihydropyridine and non-dihydropyridine. In contrast, a non-selective β-blocker, propranolol, may aggravate VSA [36]. Nitrate can relieve CAS promptly, but its role in VSA prevention is limited by tolerance and poor long-term clinical outcomes [37]. Some clinical research shows that magnesium [38], antioxidants [39,40], and Rho-kinase inhibitors [41] are also helpful for treatment of VSA. Additionally, precipitating factors for VSA should be absolutely avoided, e.g., alcohol, cigarette smoking, and propranolol [11]. Coronary intervention is not helpful for drug-refractory VSA [42] and is contraindicated in patients without angiographical CAD because of the presence of diffuse spastic characteristics in the setting of CAS [4]. Cardioverter defibrillator implantation with adequate pharmacological therapy for CAS was suggested to be an appropriate option for patients who had syncope or ventricular tachycardia or had survived hospital cardiac arrest [43]. Pharmacological treatment for VSA with calcium antagonists is suggested to be lifelong, not only because of persistent long-term spasticity of the coronary arteries [44] but also the probability of silent myocardial ischemia. Silent myocardial ischemia caused by any pathologies could be complicated by fatal or nonfatal cardiovascular events, even, as previously mentioned, ventricular arrhythmias and cardiac death. Despite the 5.5–11% recurrence rate of VSA, the long-term prognosis of VSA is good if adequate treatment is prescribed [17,45].

Summary: VSA must be diagnosed and treated correctly based on the following [46]:Angina occurs at rest and is promptly relieved by administering nitrates, but the diagnosis of VSA must further be correctly confirmed;VSA can present as stable angina, acute coronary syndrome, syncope, cardiac arrhythmias, heart failure, and sudden death;Without intracoronary testing for vasomotion, angiographically patent coronary arteries should not be interpreted as normal coronary arteries;Intracoronary provocative testing must be well prepared and undertaken based on the guidelines and should not be considered a risky procedure;Calcium antagonists are the first choice for the treatment of VSA and should be given at the right time and in the right doses.

## 3. Relation of Local and Systemic Inflammation to VSA

No single mechanism can be held responsible for the development of CAS. Some mechanisms have been proven to play a role in CAS causing VSA, i.e., allergy [47], oxidative stress [48], endothelial dysfunction [49], deficient aldehyde dehydrogenase 2 activities [50], chronic low-grade inflammation [51], magnesium deficiency [38], and hypercontraction of coronary artery smooth muscle [52]. Furthermore, age, cigarette smoking, and high-sensitivity C-reactive protein (hs-CRP) are risk factors for VSA [53]. Other factors act as inducers for VSA occurrence [11], such as physical and/or mental stress, alcohol consumption, Valsalva maneuver, hyperventilation, and other pharmacological agents, such as propranolol, ergot alkaloids, sympathomimetics and parasympathomimetics, and cocaine. Chronic low-grade inflammatory conditions seem to play the central role, interacting with each of the above-mentioned mechanisms. Although different pathophysiologies exist in VSA, the final pathway is contraction of coronary artery smooth muscle, clinically causing VSA [54]. Based on the above prior studies, it is suggested that the underlying mechanism in the development of CAS is multifactorial. The etiology of the hyperreactivity of the coronary vessels is unclear but could be related to endothelial dysfunction and the primary smooth muscle cells of the coronary vessels, which might have impaired regulatory mechanisms for vasoconstriction and vasodilation. Balances within the sympathetic and parasympathetic tone also regulate the coronaries’ flow. Since multiple factors can contribute to the development of VSA, an occurrence of VSA is variable and therefore unpredictable [44].

In 1978, Lewis and colleagues [55] described a patient who was deceased due to cardiogenic shock because of inferior wall ST-segment elevation associated with localized pericarditis. These investigators initially suggested an interaction between chronic inflammation and CAS. Subsequently, Forman et al. [56] found a VSA patient who presented with sudden death, in whom infiltrating mast cells were found at the adventitia of a spastic coronary artery. In 1988, Ferguson et al. [57] reported a 17-year-old boy who had developed two episodes of VSA following assumed acute viral myocarditis. In 1991, Iwasaki et al. [58] reported CAS in a 59-year-old male with biopsy-proven acute myocarditis. In 2008, Yilmaz et al. [59] found that CAS without CAD occurs in 70% of endomyocardial biopsy-proven PVB19 myocarditis and suggested that CAS plays an important role in the occurrence of angina pectoris in these patients. Other studies have also found intimal injury and neointimal hyperplasia with infiltrating inflammatory cells in coronary plaques or arteries in patients with VSA [60,61]. Despite a lack of angiographical evidence of coronary artery narrowing, diffuse intimal thickening in spastic arteries has been demonstrated by intracoronary ultrasound [62]. Using ^18^F-fluorodeoxyglucose positron emission tomography/computed tomography, inflammatory changes in coronary adventitia and perivascular adipose tissue were found to be associated with CAS in VSA patients [63]. Coronary perivascular ^18^F-fluorodeoxyglucose uptake decreased after prescription of a calcium antagonist in patients with VSA. Furthermore, adventitial vasa vasorum significantly increased in VSA patients, as confirmed by optical coherence tomography analysis. All the above findings suggest that local coronary inflammatory changes play a role in the early anatomical changes in the coronary arteries in CAS (Table 1), which was also suggested by Marzilli and colleagues [64]. These early anatomical changes in the coronary arteries in CAS might induce subsequent functional changes in these arteries, which might be the basis of future characteristics of spastic coronary arteries.

Increased levels of soluble intercellular adhesion molecule-1 or secretory type II phospholipase A2 have been noted in patients with VSA [65,66]. Our prior serum inflammatory biomarker studies also found increased levels of hs-CRP, interleukin-6, monocyte chemoattractant protein-1, soluble intercellular adhesion molecule-1, and soluble vascular adhesion molecule-1 in patients with VSA [51,67], indicating that systemic inflammatory changes associated with subsequent endothelial dysfunction are present in spastic coronary arteries. Endothelial dysfunction is the earliest process of atherosclerotic lesion formation [68]. Furthermore, atherosclerosis impairs the coronary arterial vasodilator function, which is an important function of endothelium [69]. Recently, we also found elevated peripheral leukocyte Rho-associated coiled-coil-containing protein kinase activity in patients with VSA [70]. Rho-associated coiled-coil-containing protein kinase activity was decreased in the VSA group after treatment with antispastic agents for 3 months. Rho-associated coiled-coil-containing protein kinase activity was independently associated with diagnosis of VSA and was found to be correlated with VSA activity. Rho-associated coiled-coil-containing protein kinase activation has been noted in association with attenuated endothelial nitric oxide synthase expression [71], increased vascular smooth muscle cell DNA synthesis and migration [72], and increased monocyte adhesion and spreading [73]. Some molecular studies in the porcine model with interleukin-1 beta showed that the expressions of Rho-kinase mRNA and RhoA mRNA were increased in the spastic coronary segment as compared with the control coronary segment [74]. Using a Rho-kinase inhibitor, Y-27632, not only inhibited serotonin-induced vascular smooth muscle hypercontraction but also accentuated myosin binding subunit phosphorylation [75]. The above molecular studies indicate that Rho-kinase is upregulated at the spastic site and causes vascular smooth muscle hypercontraction. Therefore, the pathogenesis of VSA could be a combination and interplay of endothelial dysfunction, systemic inflammation, and smooth muscle hypercontraction. Our series of CAS studies and other prior CAS studies do not include patients with obstructive CAD; low-grade systemic inflammation is present in these VSA patients, similar to that found in obstructive CAD patients. Therefore, it is reasonable to infer that coronary arteries undergoing CAS are not normal, and that systemic inflammatory status exists in VSA, because abnormal endothelial function and diffuse intimal thickening causing inadequate nitric oxide synthesis is observed in these patients. 

In 1991, Kounis et al. [47] postulated a concept of allergic angina based on observing an acute allergic condition associated with acute coronary syndromes. They then suggested that histamine, the main amine during allergy, could induce CAS manifested as VSA or acute myocardial infarction. Subsequently, they modified their understanding of the Kounis syndrome towards mast cell activation [76], further making an argument for allergic inflammatory-response-induced CAS.1 There are three variants of the Kounis syndrome [76], i.e., Type I: allergic VSA due to endothelial dysfunction in patients without underlying CAD, Type II: an allergic reaction causing CAS or plaque erosion in patients with underlying asymptomatic CAD, and Type III: an allergic CAS in the setting of coronary thrombosis, including stent thrombosis. Our prior case report demonstrated that type I Kounis syndrome occurred in a 45-year-old sigmoid cancer patient who had drug-allergic VSA with the chemotherapy agent oxaliplatin [77]. Because treatment strategies for Kounis syndrome and asthma are not exactly the same as for pure VSA, knowledge of individual hypersensitivity is required. 

Using the National Health Insurance Research Database, we also noticed that asthma is independently associated with new-onset VSA (odds ration = 1.85) [78], providing further evidence of the interplay between allergic reaction and CAS. In this study, the risk of new-onset VSA was higher in prior steroid users irrespective of the oral (odds ratio = 1.22) or inhaled route (odds ratio = 1.89). Further analysis showed that the prevalence of asthma in VSA patients (4.4%) was the highest, followed by patients who had VSA associated with atherosclerotic coronary artery disease (2.6%) and atherosclerotic coronary artery disease treated by coronary intervention (1.8%). These results further indicate that an interplay exists between the bronchial spasm of asthma and the CAS of VSA. Inflammation can contribute to the occurrence of asthma [79]. As a result, the inflammatory process plays an important role in the occurrence of bronchial spasm and CAS. 

Smoking is an important association factor for VSA [80]. Our investigation [81] reported an odds ratio of 2.58, similar to a prior CAS investigation’s 2.41 [76]. A synergistic interaction between smoking and hs-CRP was further identified in the study [81]. Among smokers, the interaction was linear and monotonic. In non-smokers, a threshold effect of hs-CRP was observed on VSA. After adjusting for hs-CRP as a confounder in analyzing the impact of smoking on VSA development, a decreased odds ratio was found, suggesting hs-CRP as an important covariate of VSA. Furthermore, we found that the relation of hs-CRP to VSA is different between genders [53]. A non-threshold model for male patients and a threshold model for female patients can be interpreted as more male smokers (lifestyle) and older smokers (induction time) contributing to the natural history of VSA development. Interestingly, hypertension was found to be negatively associated with VSA [82], suggesting that VSA is different from coronary atherosclerosis in terms of pathogenesis. Recently, our cellular study [83] also noted that elevated levels of monocytic interleukin-6 and α7 nicotinic acetylcholine receptor mRNA expression and protein production are related to the interaction between nicotine and C-reactive protein. This effect is positive on the occurrence of CAS. Another big data analysis using the National Health Insurance Research Database found that anxiety and depression diagnosis are risk factors for VSA [84]. Patients with anxiety and depression have a higher risk of new-onset CAS compared with new-onset atherosclerotic coronary artery disease (odds ratios = 2.29 and 1.34, respectively). Further analysis found that a stronger risk association is noted when comparing CAS with a control group without atherosclerotic coronary artery disease or CAS (odds ratios = 5.20 and 1.98, respectively). In this study, there was no gender difference in the association of anxiety and depression with CAS. An elevated inflammatory condition in patients with depression and anxiety with potential causality has been documented in United Kingdom Biobank and Netherlands Study of Depression and Anxiety cohorts [85]. Using the National Health Insurance Research Database, we noted that CAS is associated with incident diabetes irrespective of gender, indicating a link between the inflammation of VSA and the insulin resistance of incident diabetes [86]. Insulin resistance is a central marker of metabolic syndrome, and its positive association with VSA has been identified [87,88]. Inflammation exists in the state of insulin resistance [89]. Insulin resistance is associated with compensatory hyperinsulinemia, which further causes endothelial dysfunction [90]. However, VSA does not occur in every patient with endothelial dysfunction [25]. A pathological phenomenon does not necessarily lead to clinical disease. Therefore, an association between systemic inflammation and VSA (Table 1) is further suggested [91,92,93].

## 4. Conclusions

VSA is a potentially lethal medical condition with multiple presentations. A detailed medical history and a follow-up series of electrocardiographic ST-segment and T-wave changes are the bases for diagnosing VSA. To identify the underlying cause of angina, especially in patients with no obstructive coronary arteries, is crucial in the primary coronary interventional era. Only correct diagnosis can lead to correct treatment. With the advancement of medical diagnostic imaging capabilities, discovery of the pathogenesis of CAS has become possible. Local and systemic inflammation in association with VSA is increasingly being recognized. Therefore, effective treatment strategies to decrease inflammation are worthy of further investigation.

## Figures and Tables

**Table 1 medicina-59-00318-t001:** Relationship between inflammation and vasospastic angina.

Inflammatory Status	Comments
Local	
Pericarditis	Case report
Mast cells infiltration in adventitia of spastic coronary artery	Case report
Acute viral myocarditis	Biopsy-proven
Intimal injury and intimal hyperplasia	Histology
Diffuse intimal thickening	Intravascular ultrasound
Inflammation of Coronary adventitia and perivascular adipose tissue	Positron emission tomography
Adventitial vasa vasorum increase	Optical coherence tomography
Systemic	
Elevated circulatory inflammatory and adhesion markers	Plasma/serum studies
Elevated peripheral leukocyte ROCK activity	Protein expression studies
Kounis syndrome	Clinical disease entity association
Asthma association	Clinical disease entity association
Anxiety/depression association	Clinical disease entity association
Insulin resistance	Clinical disease entity association
Cigarette smoking association	Clinical risk factor

## Data Availability

Not applicable.

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
