# Peer review of "Relationship between Inflammation and Vasospastic Angina"

_medicina, 2023, doi:10.3390/medicina59020318_

Round 1

Reviewer 1 Report

Dear authors,

The manuscript "Relationship between Inflammation and Vasospastic Angina" is well-written with much scientific information. 

The underlying mechanism in the development of coronary artery spasm is multifactorial. The etiology of the hyperreactivity of the coronary vessels is unclear but could be related to endothelial dysfunction and primary smooth muscle cells of the coronary vessels that might have impaired regulatory mechanism for vasoconstriction and vasodilation. Balances within the sympathetic and parasympathetic tone also regulate the coronaries’ flow.

Inflammation is one of the essential etiology factor that provokes VAS. On the other hand, cigarette smoking and metabolic disorder such as insulin resistance are also significant factors.

The authors have not mentioned insulin resistance.

Additionally, in patients with VAS a noninvasive stress test can be performed, such as a hyperventilation test and stress echocardiography. So, the authors should mention them in the manuscript.

Author Response

Reviewer 1:

  1. The underlying mechanism in the development of coronary artery spasm is multifactorial. The etiology of the hyperreactivity of the coronary vessels is unclear but could be related to endothelial dysfunction and primary smooth muscle cells of the coronary vessels that might have impaired regulatory mechanisms for vasoconstriction and vasodilation. Balances within the sympathetic and parasympathetic tone also regulate the coronaries’ flow.

Response: We appreciate the positive comments. The above comments have been added to the revised manuscript (page 10, lines 7-13).

  1. Inflammation is one of the essential etiology factors that provoke VAS. On the other hand, cigarette smoking and metabolic disorder such as insulin resistance are also significant factors. The authors have not mentioned insulin resistance.

Response: We appreciate the important comments. Using the National Health Insurance Research Database, we also noted that CAS is associated with incident diabetes irrespective of gender indicating a link between inflammation of VSA and insulin resistance of incident diabetes [82]. Insulin resistance is a central marker of metabolic syndrome and its positive association with VSA has been identified [83, 84]. Inflammation exists in the status of insulin resistance [85]. Insulin resistance is associated with compensatory hyperinsulinemia, which further causes endothelial dysfunction [86]. However, VSA does not occur in every patient with endothelial dysfunction [22]. The above statements have been added in the revised manuscript (page 15, line 13 to page 16, line 4).

  1. Additionally, in patients with VAS a noninvasive stress test can be performed, such as a hyperventilation test and stress echocardiography. So, the authors should mention them in the manuscript.

Response: We appreciate the comments. In addition to intracoronary stress testing, other non-invasive stress modalities have been used to diagnose VSA such as hyperventilation [28] and stress echocardiography using either cold-pressor testing or intravenous ergonovine testing [29, 30]. These non-invasive modalities are alternative methods in diagnosing VSA if there are contraindications to undergo invasive coronary angiography and are suggested to be performed by experienced physicians. The above statements have been added in the revised manuscript (page 7, lines 6-10).

Reviewer 2 Report

Relationship between Inflammation and Vasospastic Angina
Hung presents a comprehensive overview of the topic of vasospastic angina, including historical perspective, diagnosis, and special focus on possible pathophysiologic mechanisms.

General comments:
The manuscript is well written. I didn't notice any scientific problems.
A brief section on incidence in general or in different populations (Asians vs. Westerners) might help to capture the magnitude of the problem, but this is only a suggestion.

Author Response

  1. A brief section on incidence in general or in different populations (Asians vs. Westerners) might help to capture the magnitude of the problem, but this is only a suggestion.

Response: We appreciate the comments. In the 1990s, VSA was reported to have a higher prevalence in the Japanese population compared with the western population [14, 15]. Subsequently, the prevalence rates of VSA in Taiwan and Korea were reported to be similar to Japan [16, 17]. However, recent European studies found that VSA is not as uncommon as previously thought in white patients with angina pectoris and myocardial ischemia with unobstructive coronary arteries [18, 19]. There was no difference in patterns of CAS in these studies; however, higher proportions of males and smoking history were noted in the Japanese population [18]. Therefore, ethnic heterogeneities in VSA remain further research. The above statements have been added in the revised manuscript (page 4, line 16 to page 5, line 7).